# SFT-KD-Recon: Learning a Student-friendly Teacher for Knowledge Distillation in Magnetic Resonance Image Reconstruction

**Matcha Naga Gayathri**[*1]             EE21S048@SMAIL.IITM.AC.IN
**Sriprabha Ramanarayanan**[*1,2]          SRIPRABHA.R@HTIC.IITM.AC.IN
**Mohammad Al Fahim**[1]              EE21S050@SMAIL.IITM.AC.IN
**Rahul G S**[2]                   RAHUL.G.S@HTIC.IITM.AC.IN
**Keerthi Ram**[2]                  KEERTHI@HTIC.IITM.AC.IN
**Mohanasankar Sivaprakasam** [1,2]         MOHAN@EE.IITM.AC.IN
[1] *Indian Institute of Technology, Madras*
[2] *HealthCare Technology Innovation Center*

**Editors:** Accepted for publication at MIDL 2023

## Abstract

Deep cascaded architectures for magnetic resonance imaging (MRI) acceleration have shown remarkable success in providing high-quality reconstruction. However, as the number of cascades increases, the improvements in reconstruction tend to become marginal, indicating possible excess model capacity. Knowledge distillation (KD) is an emerging technique to compress these models, in which a trained deep 'teacher' network is used to distill knowledge to a smaller 'student' network, such that the student learns to mimic the behavior of the teacher. Most KD methods focus on effectively training the student with a pre-trained teacher that is unaware of the student model. We propose SFT-KD-Recon, a student-friendly teacher training approach along with the student as a prior step to KD to make the teacher aware of the student's structure and capacity and enable aligning the teacher's representations with the student. In SFT, the teacher is jointly trained with the unfolded branch configurations of the student blocks using three loss terms - teacher-reconstruction loss, student-reconstruction loss, and teacher-student imitation loss, followed by KD of the student. We perform extensive experiments for MRI acceleration in 4x and 5x under-sampling, on the brain and cardiac datasets on five KD methods using the proposed approach as a prior step. We consider the DC-CNN architecture and setup teacher as D5C5 (141765 parameters), and student as D3C5 (49285 parameters) denoting 2.87:1 compression. Results show that (i) our approach consistently improves the KD methods with improved reconstruction performance and image quality, and (ii) the student distilled using our approach is competitive with the teacher, with the performance gap reduced from 0.53 dB to 0.03 dB.

**Keywords:** Knowledge Distillation (KD), Student-Friendly Teacher KD for reconstruction (SFT-KD-Recon), MRI, deep cascaded convolutional neural networks (DC-CNN).

## 1. Introduction

Modern deep neural networks have achieved outstanding performance in various medical imaging tasks such as image reconstruction, super-resolution, and object detection. Specifically in Magnetic Resonance Imaging (MRI) acceleration, the top five performing methods

---

* Contributed equally

of fastMRI were implemented by using cascades of deep learning (DL) models (Muckley et al., 2021), with greater depth and complexity yielding higher output fidelity. However, the gain in performance per added cascade is not linear (Ramanarayanan et al., 2020a) over discrete increments in model size but rather becomes marginal asymptotically, implying possible excess capacity and potential compressibility of cascade models like DC-CNN (Schlemper et al., 2017) and DC-UNet (Sun et al., 2019).

The field of model compression research addresses the problem of achieving the least network size with minimum reduction in performance. To obtain lightweight models, there exist recent techniques such as (1) model pruning [(Li et al., 2016), (He et al., 2017)], where the model weights are sparsified to minimize redundancies, (2) lightweight network design [(Howard et al., 2017), (Zhang et al., 2018a)], such as substituting convolutions with separable convolutions, and (3) knowledge distillation (KD) methods [(Hinton et al., 2015), (Romero et al., 2014), (Yim et al., 2017b), (Gao et al., 2018)], where a trained deep 'teacher' network is used to distill representations to a smaller 'student' network, and the student learns to mimic the behavior of the teacher network, retaining the structure (Fig. 1a).

KD is an interesting method, as besides distilling from a pre-trained teacher network (Hinton et al., 2015), the training framework can accommodate the training of the teacher model along with the student. Such approaches can render the teacher aware of the student's structure and capacity, and offer scope for aligning representations with the student's layer features. A real-world analogy is neuro-linguistic programming, wherein educators understand what motivates students and orient the teaching to suit them (Tosey and Mathison, 2010). Student-Friendly Teacher (SFT) training (Park et al., 2021) is a recent method that introduces a prior step of jointly training the teacher along with the student (i.e. student-aware training), followed by routine feature-based knowledge distillation. This approach uses modular block-structured architectures for both teacher and student and augments the student branches to the teacher during student-aware teacher training. This has been shown to improve top-1 classification accuracy in standard image classification datasets using ResNet-50 (teacher) and ResNet-34 (student) with a compression factor of 32 %.

For MRI reconstruction, we consider the deep cascaded convolutional neural network (DC-CNN) owing to its block-structured configuration and good-quality reconstruction. For KD, we consider the DC-CNN with deeper blocks as the teacher and shallower blocks as the student. We perform SFT training by enabling the teacher to interact with the student block-wise. This student-oriented approach enables the low-level features of the student to match with the corresponding features of the teacher, improving both the teacher and the student. Furthermore, we effectively reuse the mutual knowledge (Zhang et al., 2018b) learned by the student for the subsequent distillation process. Hence, we have chosen the SFT-KD framework to ensure consistency between the teacher and the student for optimal knowledge distillation in MRI reconstruction. In SFT, the ResNet units need to be configured as blocks, wherein the number of Resnet blocks might be a hyperparameter and each block output is in the residual feature domain. In ours, the deep cascaded MRI reconstruction CNNs with interleaved data fidelity blocks are inherently block-structured with each block output in the image domain (Figure 2). Our contributions are as follows:

1. We propose SFT-KD-Recon, a Student-Friendly Teacher learning framework for enhancing knowledge distillation for MRI restoration tasks. The proposed framework enables knowledge distillation to be oriented to the student network.

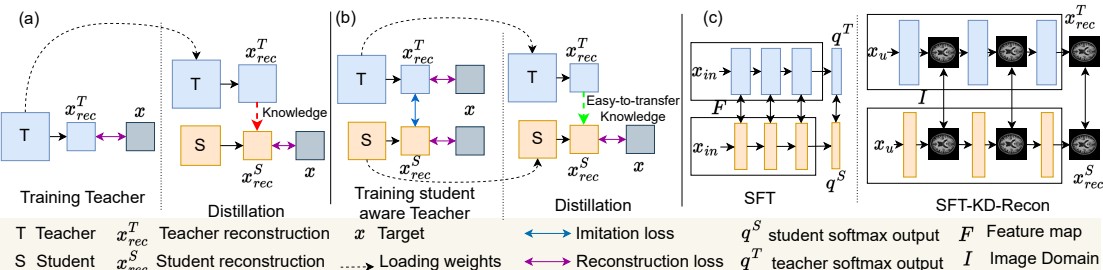

Figure 1: Comparison between the standard KD and SFT-KD-Recon. (a) The standard KD trains teacher alone and distills knowledge to student. (b) SFT-KD-Recon trains the teacher along with the student branches and then distills effective knowledge to student. (c) SFT Vs SFT-KD-Recon, the former learns in the feature domain via residual CNN while the latter learns in the image domain via image domain CNN.

2. The proposed framework improves the teacher and provides an initialization to the student layers for the subsequent stage of distillation. Our method is straightforward and can be easily plugged into the standard KD methods for high fidelity.

3. We demonstrate the effectiveness of SFT-KD-Recon on various KD methods for MRI reconstruction and super-resolution tasks. Experiments reveal (i) the superiority of the proposed student-friendly teacher over conventional teacher models, and (ii) the consistency of SFT-KD-Recon in improving the student accuracy across five KD methods for image reconstruction and super-resolution.

## 2. Related work

**MRI reconstruction** using cascaded image domain architectures like deep cascaded CNN (DC-CNN) (Schlemper et al., 2017), dilated networks (Sun et al., 2018), attention mechanism (Huang et al., 2019), dense connections (Wu et al., 2018), and adaptive reconstruction (Ramanarayanan et al., 2020b) have shown promising performance for various anatomies, contrasts and acceleration factors for reconstruction. To demonstrate the efficacy of our method we have chosen the foremost and simplest deep cascaded architecture, DC-CNN.

**Knowledge distillation** is first introduced for neural network model compression (Hinton et al., 2015). KD methods for classification tasks include, (i) FitNets (Romero et al., 2014) which uses the pre-trained teacher's hint layer and student's guided layer for distillation, (ii) Flow of solution procedure (FSP) (Yim et al., 2017a), wherein the student preserves the pairwise similarities with the teacher in the representation space, (iii) Similarity-preserving (SP) KD (Tung and Mori, 2019) wherein the student mimics the similarity map of the intermediate layers of the teacher, and (iv) Correlation Congruence (CC) (Peng et al., 2019) wherein the information at the input instance level and the correlation between instances are transferred to the student. Unlike KD for classification tasks wherein knowledge refers to the softened probabilities or sample relationships, for regression tasks, KD remains less explored. For MRI reconstruction, both universal under-sampled reconstruction (Liu et al., 2021) and KD-MRI (Murugesan et al., 2020) use Attention transfer (AT) (Zagoruyko and Komodakis, 2016) which distills the response patterns in the teacher to the student feature maps. A few other previous works for image regression tasks include FAKD (super-resolution) (He et al., 2020), U-Net KD (denoising) (Chen et al., 2021), collaborative KD

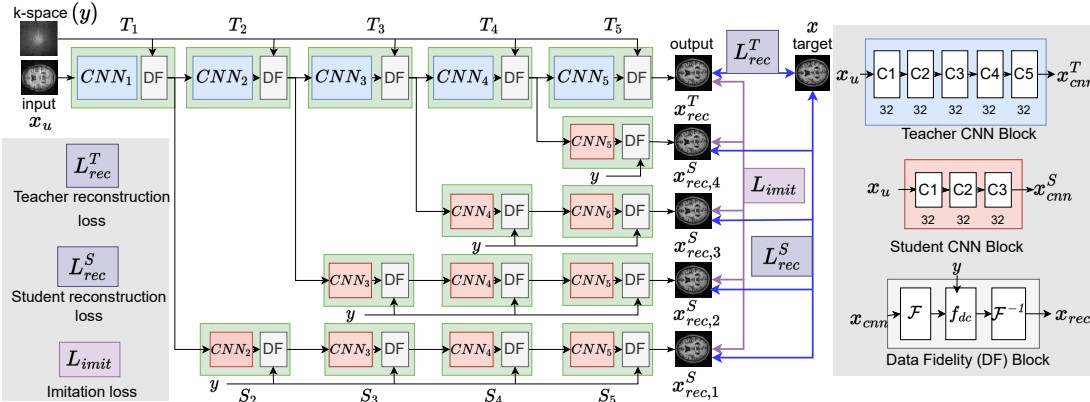

Figure 2: Student-Friendly training of the teacher. The teacher DC-CNN has five blocks, each having CNN with five convolution layers and DF layer, and the student DC-CNN has five blocks, each having three convolution layers and a DF layer. The teacher is trained with three loss terms - $L_{rec}^T$, $L_{rec}^S$ (blue arrows), and $L_{imit}$ (violet arrows). Note that all the blocks of the student learn initial weights except the first block during SFT training.

(style transfer) (Wang et al., 2020) and deep pose regression network (Saputra et al., 2019). Similar to these networks, our work focuses on model compression, and learns the feature similarity with the teacher to improve the student's fidelity.

## 3. Methodology

### 3.1. Problem formulation for deep learning based MRI reconstruction

The deep-learning-based MRI reconstruction can be formulated as an optimization problem (Huang et al., 2019) and is given by:

$$\min_x \|x - f_{cnn}(x_u \mid \theta)\|_2^2 + \lambda \|\mathcal{A}x - y\|_2^2 \tag{1}$$

where $x \in \mathbb{C}^N$ denotes the desired image, $y \in \mathbb{C}^M$ is the under-sampled k-space measurements. Here $\mathcal{A} : \mathbb{C}^N \to \mathbb{C}^M$ represents the forward operator of the MRI acquisition and is given by $\mathcal{A} = \mathcal{M} \circ \mathcal{F}(x)$ where $\circ$ indicate Hadamard product. $\mathcal{M}$ is the under-sampling mask and $\mathcal{F}(.)$ is the 2D Fourier transform. Here, $f_{cnn}$ is the CNN that learns the mapping between the zero filled (ZF) or under-sampled image $x_u$ and the fully sampled image $x$ and is parameterized by $\theta$. The data fidelity (DF) operation in the k-space domain is performed after each CNN block to ensure consistency with the acquired k-space and is given by:

$$\hat{X}_{rec} = \begin{cases} \hat{X}_{cnn}(k) & k \notin \Omega \\ \frac{\hat{X}_{cnn}(k) + \lambda \hat{X}_u(k)}{1+\lambda} & k \in \Omega, \lambda \to \infty \end{cases} \tag{2}$$

where $x_{cnn} = f_{cnn}(x_u \mid \theta)$, $\hat{X}_{cnn} = \mathcal{F}(x_{cnn})$, $\hat{X}_u = \mathcal{F}(x_u)$, $x_{rec} = \mathcal{F}^{-1}(\hat{X}_{rec})$, $\Omega$ is the index set of the acquired k-space measurements and $\lambda$ is the data fidelity weight.

### 3.2. Proposed Student-friendly Teacher-KD Framework

The SFT approach aims to train the teacher model collaboratively with the student, prior to distillation. During teacher training, the teacher learns along with the student branches

to obtain representations tailored to the student. Figure 2 shows that SFT training consists of a teacher network along with multiple branches from the student network. Let $\{T_n\}_{n=1}^N$ and $\{S_n\}_{n=1}^N$ denote the blocks in the teacher and the student respectively. Here $N$ denotes the number of blocks in the two networks. The figure shows a case with $N = 5$. Here, $T_1$ block is associated with $S_2 - S_3 - S_4 - S_5$ branch, $T_2$ is associated with $S_3$ to $S_5$, and so on. By infusing student intermediate branches with the corresponding blocks of the teacher, each teacher block improves its knowledge based on the respective student branch and also the knowledge of the subsequent teacher blocks. For instance, T1 learns according to the $S_2$ to $S_5$ branch and $T_2$ to $T_5$ blocks. Each student branch performs image-to-image mapping which is essential for reconstruction, unlike SFT, which learns in the feature domain. Our three loss terms are:

**(1) *Teacher-target reconstruction loss:*** The $L1$ loss between the teacher's prediction $x_{rec}^T$, and the target $x$, given by

$$L_{rec}^T = \left\| x - x_{rec}^T \right\|_1 \tag{3}$$

**(2) *Student-target reconstruction loss:*** The average $L1$ loss taken over each student branch output $x_{rec,i}^S$, $i = 1, 2, ..N - 1$ and the target, given by

$$L_{rec}^S = \frac{1}{N-1} \sum_{i=1}^{N-1} \left\| x - x_{rec,i}^S \right\|_1 \tag{4}$$

**(3) *Student-teacher imitation loss:*** The $L1$ loss that minimizes the inconsistency between the predictions of the teacher and the student (branch-wise), given by

$$L_{imit} = \frac{1}{N-1} \sum_{i=1}^{N-1} \left\| x_{rec}^T - x_{rec,i}^S \right\|_1 \tag{5}$$

The overall training process is illustrated in Algorithm 1

---

**Algorithm 1:** SFT-KD-Recon procedure:

- Step 1: Train the teacher DC-CNN, $f_{cnn}^T$ parameterized by $\theta^T$, along with multiple student branches with loss $L_{SFTN} = L_{rec}^T + L_{rec}^S + L_{imit}$.

- Step 2: Load the student blocks weights $\theta^S$ obtained during SFT training in Step 1, and train the student network, $f_{cnn}^S$ using reconstruction loss $L_{rec}^S = \left\| x - x_{rec}^S \right\|_1$ and distillation loss (depending upon the KD method) between teacher and student.

---

## 4. Experiments and Results

### 4.1. Dataset Description and Evaluation metrics

We have used two publicly available datasets to evaluate the proposed method for MRI reconstruction. (1) **The Cardiac MRI dataset**, released as part of the Automated Cardiac Diagnosis Challenge (ACDC) (Bernard et al., 2018), consists of 150 patient records

Table 1: Comparison of our framework with standard KD framework for MRI Reconstruction on MRBrainS and cardiac datasets. In all the KD methods, the student distilled from the SFT-KD-Recon outperforms the ones distilled from the standard teacher.

| Dataset | Model | 4x | | 5x | |
|---------|-------|-----|------|-----|------|
| | | Std-KD | SFT-KD-Recon | Std-KD | SFT-KD-Recon |
| | | PSNR/SSIM | PSNR/SSIM | PSNR/SSIM | PSNR/SSIM |
| MRBrainS | ZF | 31.38/0.6651 | - | 29.93/0.6304 | - |
| | Teacher | 40.05/0.9785 | **40.23/0.9799** | 39.11/0.9715 | **39.16/0.972** |
| | Student | 39.49/0.9759 | - | 38.49/0.9663 | - |
| | Fitnets | 39.52/0.9762 | **40.08/0.9790** | 38.70/0.9681 | **38.95/0.9699** |
| | AT | 39.76/0.9769 | **40.07/0.9789** | 38.85/0.9691 | **38.92/0.9699** |
| | CC | 39.64/0.9762 | **40.06/0.9789** | 38.60/0.9675 | **38.86/0.9691** |
| | FSP | 39.45/0.9756 | **40.01/0.9786** | 38.32/0.9643 | **38.82/0.9692** |
| | SP | 39.53/0.9762 | **39.93/0.9782** | 38.46/0.9661 | **38.81/0.9692** |
| Cardiac | ZF | 24.27/0.6996 | - | 23.82/0.6742 | - |
| | Teacher | 32.15/0.9108 | **32.26/0.9126** | 31.25/0.8964 | **31.33/0.8968** |
| | Student | 31.68/0.9013 | - | 30.59 /0.8826 | - |
| | AT | 31.95/0.9060 | **32.03/0.9070** | 30.88/0.8879 | **30.93/0.8884** |
| | Fitnets | 31.90/0.9050 | **31.95/0.9067** | 30.59/0.8811 | **30.86/0.8871** |
| | CC | 31.83/0.9049 | **31.96/0.9070** | 30.73/0.8859 | **30.94/0.8889** |
| | FSP | 31.71/0.9024 | **31.91/0.9060** | 30.51/0.8817 | **30.82/0.8862** |
| | SP | 31.58/0.9004 | **31.91/0.9060** | 30.73/0.8860 | **30.83/0.8865** |

(1841 slices) for training and 50 patient records (1076 slices) for validation. Each slice is cropped to its central $150 \times 150$ region. (2) **The Brain MRI dataset**, MRBrainS Dataset (Mendrik et al., 2015) consists of 7 volumes of T1 MRI split into 5 volumes (240 slices) for training and the remaining (96 slices) for validation. Each slice is of size $240 \times 240$. The under-sampled k-space and under-sampled images are retrospectively obtained using fixed Cartesian under-sampling masks for 4x and 5x acceleration factors (Ramanarayanan et al., 2020a). We use Peak Signal-to-Noise Ratio (PSNR) and Structural Similarity Index (SSIM) metrics as our evaluation metrics. Wilcoxon signed rank test with an alpha of 0.05 is used to assess statistical significance.

## 4.2. Implementation Details

For KD, we use DC-CNN configurations $D5C5$ as the teacher and $D3C5$ as the student with a compression factor of 65%. Here $D3C5$ means each cascade block has 3 convolution layers and there are 5 cascade blocks. Each layer in the two networks has 32 channels with ReLU activations and 3x3 filters. The cascaded network have alternating CNN blocks and DF units. We have chosen five KD methods to demonstrate the effectiveness of our approach, namely AT (Attention Transfer) (Zagoruyko and Komodakis, 2016), Fitnets (Romero et al., 2014), Similarity Procedure (SP) (Tung and Mori, 2019), Flow of Solution Procedure (FSP) (Yim et al., 2017a), and CC (Correlation congruence) (Peng et al., 2019).

Models are implemented in PyTorch (v1.12) on a 24GB RTX 3090 GPU. For every step mentioned in the training algorithm, models are trained for 150 epochs using the Adam optimizer, with a learning rate of $1e^{-3}$. Code for our proposed method is available at SFT-KD-Recon repository [1].

---

1. https://github.com/GayathriMatcha/SFT-KD-Recon

### 4.3. Results and discussion

We compare our SFT-KD-Recon (Student trained using our procedure) with Std-KD (standard KD i.e. student trained using pre-trained teacher (Hinton et al., 2015)), student, and teacher in 4x and 5x setup on cardiac and brain datasets. Table 1 provides the quantitative analysis of the reconstruction performance of the proposed SFT approach applied to five different KD methods. From the table, our observations are as follows. (i) Student-friendly teacher performs consistently better than the conventional teacher with the highest improvement margin of 0.15 dB in PSNR and 0.002 in SSIM. (ii) SFT-KD-Recon is consistently better than Std-KD in the five KD methods. (iii) In PSNR, the performance drop in the student relative to the teacher is 0.56 dB without KD. By using Std-KD, this gap is 0.53 dB. With the proposed SFT-KD-Recon, the gap is significantly minimized to 0.03 dB.

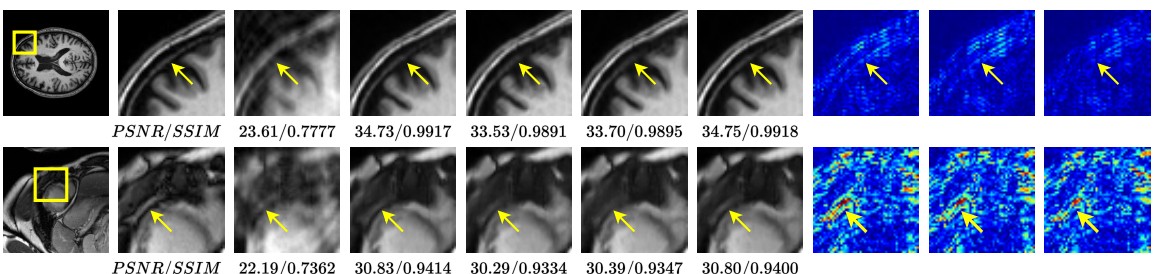

Figure 3: Visual results (from left to right): target, target inset, ZF, teacher, student, Std-KD, SFT-KD-Recon, student residue, Std-KD residue, SFT-KD-Recon residue with respect to the target, for the brain (top) and cardiac (bottom) with 4x acceleration. We note that in addition to lower reconstruction errors, the SFT-KD distilled student is able to retain finer structures better when compared to the student and Std-KD output.

We note that during SFT training, multiple configurations of the students ((i) $S_5$ alone, (ii) $S_4$ and $S_5$, (iii) $S_3$, $S_4$ and $S_5$, and so on) are collaboratively involved. The teacher and each student configuration are primarily directed by a supervised learning loss. Each configuration of student branch is a sub-network that steers towards a common objective. Each student sub-network starts with different initialization and provides predictions that serve as extra information for the teacher to optimize to a more robust set of features (Zhang et al., 2018b). This joint learning of the teacher makes it oriented toward student and transfers effective knowledge during the distillation stage as shown in our experiments. Interestingly, the distilled student performs as good as the original teacher in three KD methods, namely AT, Fitnets, and CC for 4x acceleration.

Our PSNR / SSIM results for AT which has given best KD results are: Teacher: 40.05/0.9785, Student: 39.49/0.9759, KD: 39.76/0.9769, SFT-KD-Recon (random initializations): 40.01/0.9786, and with initializations: 40.07/0.9789. This shows that the weights learned by student branches during SFT training are helpful to provide better initializations to the student. Comparing conventional SFT and ours using MSE loss and AT as KD method shows that SFT-KD-Recon outperforms conventional SFT (Table 2). Although, in general, feature domain learning is important for KD, image domain learning is crucial both for the SFT training and the KD, using cascaded MRI reconstruction networks.

Table 2: Comparison of SFT and SFT-KD-Recon for AT distillation method on MRBrainS dataset and 4x acceleration. For SFT, the classification task loss terms are replaced with MSE loss for reconstruction. Our framework which trains in the image domain ensures better reconstruction fidelity than feature domain learning.

| Dataset | SFT training setup | | | SFT-KD-Recon training setup | | |
|---|---|---|---|---|---|---|
| | Training done with MSE loss in place of CE/KL loss | | | Training done in image domain with MSE loss | | |
| | **Model** | **PSNR** | **SSIM** | **Model** | **PSNR** | **SSIM** |
| MRBrainS, 4x | Teacher | $34.06 \pm 0.7114$ | $0.9291 \pm 0.007151$ | Teacher | $40.05 \pm 2.023$ | $0.9785 \pm 0.00655$ |
| | SF Teacher | $34.06 \pm 0.6263$ | $0.9296 \pm 0.005212$ | SFT-KD-Recon Teacher | $40.23 \pm 2.081$ | $0.9799 \pm 0.00636$ |
| | Student | $33.48 \pm 0.4628$ | $0.9199 \pm 0.0051$ | Student | $39.49 \pm 1.778$ | $0.9759 \pm 0.005949$ |
| | KD | $32.50 \pm 0.4323$ | $0.9156 \pm 0.006479$ | KD | $39.76 \pm 1.899$ | $0.9769 \pm 0.006172$ |
| | SFT | $\mathbf{33.50 \pm 0.4232}$ | $\mathbf{0.9205 \pm 0.005292}$ | SFT-KD-Recon | $\mathbf{40.07 \pm 1.983}$ | $\mathbf{0.9789 \pm 0.0062}$ |

Figure 4: (a) SSIM Box plots of KD, SFT-KD-Recon with respect to teacher and student across the brain and cardiac datasets for 4x and 5x acceleration. (b) Reconstruction loss of teacher, student, SFT-Teacher, KD, SFT-KD-Recon on the validation set for the cardiac dataset, 4x acceleration. KD and SFT-KD-Recon use AT as the distillation method.

Figure 3 shows the visual results comparing the target image, ZF, teacher, student, KD, and SFT-KD-Recon for the brain and cardiac respectively. Our observations are (i) SFT-Recon exhibits better structure recovery as compared to KD. The arrows pointing to the middle frontal gyrus region of the brain show that our method can recover details better than student and the Std-KD student. In the cardiac image, the cardiac mass pointed by the arrows for student and KD is more faded when compared to our approach. The box plots in Figure 4 show better results for our proposed approach (higher SSIM value (green) in figure 4(a) and lower reconstruction loss (green) in (b)) than student and KD. For our proposed method, the metrics are statistically significant with $p < 0.05$.

## 5. Conclusion

In this work, we introduced SFT-KD-Recon, a student-friendly teacher training approach for improved knowledge distillation in the MRI restoration tasks. In the proposed approach, the teacher network is learned along with the student in a block-structured manner to improve the accuracy of the student during KD. Extensive experimentation for MRI reconstruction and super-resolution tasks on the brain and cardiac datasets using various KD methods show that the proposed SFT training can align the teacher to the student and can significantly improve the performance of knowledge distillation in MRI.

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

## Appendix A. MRI Reconstruction

### A.0.1. Performance Analysis of SFT-KD

We show the efficacy of our approach using two other metrics, (1) High-Frequency Error Norm (HFN) (Fujita et al., 2020) and (2) Visual Information Fidelity (VIF) (Han et al., 2013). HFN quantifies the quality of reconstruction of edges and fine features of MRI. VIF is chosen to assess the quality of MRI reconstruction with radiologist perception. As shown in Table 3, it is evident that our approach improves student performance in terms of these metrics also. We obtained the best improvement for the fitnets method of around 1.3% improvement in VIF score. Note that the HFN and VIF scores of student distilled with our method are closer to the teacher and very much better than student and KD-distilled student.

Table 3: HFN (lower is better) and VIF (higher is better) score of KD, SFT-KD-Recon with respect to teacher and student for five KD methods of MRBrainS dataset for 4x acceleration factor. Apart from PSNR and SSIM, we can notice a consistent improvement in other metrics like HFN and VIF scores for a distilled student with our approach.

| Dataset | Model | 4x | |
|---|---|---|---|
| | | **Std. KD** | **SFT-KD-Recon** |
| | | HFN/VIF | HFN/VIF |
| MRBrainS | Teacher | 0.2802/ 0.9146 | **0.2769/0.9223** |
| | Student | 0.3266/0.9037 | **-** |
| | Fitnets | 0.3204/0.9012 | **0.2816/0.9166** |
| | AT | 0.3023/0.9103 | **0.2825/0.9141** |
| | CC | 0.3081/0.9078 | **0.2833/0.9155** |
| | FSP | 0.3155/0.8989 | **0.2848/0.9169** |
| | SP | 0.3187/0.9023 | **0.2915/0.9170** |

### A.0.2. COMPARISON OF SFT AND SFT-KD-RECON

To compare SFT and SFT-KD-Recon, we just took the original SFT with only residual layers and no intermediate image outputs and replaced CE/KL with MSE loss. We preserved the number of CNN layers in SFT and SFT-KD-Recon between the teacher (25 convolution layers) and the student (15 layers). Table 4 shows that SFT constitutes a set of models - teacher (25 layer CNN), student (15 layer CNN), KD (15 layer CNN), and SFT-KD-Recon (15 layer CNN) with the underlying architecture having only residual layers. Similarly, SFT-KD-Recon constitutes another set of models - teacher (D5C5), student (D3C5), KD (D3C5), SFT-KD-Recon (D3C5) with intermediate data fidelity units and image outputs. We note that the SFT-KD-Recon training setting improves the teacher and the AT distillation step significantly better than models trained in a conventional SFT setting. Figure 5 shows quantitative analysis of teacher, student, Std. KD, SFT student in both SFT and SFT-KD-Recon training settings. Figure 6 shows the comparison of the residual feature attention maps of the output layer in each cascade of the student trained using SFT and SFT-KD-Recon training settings. The residual feature maps are taken with respect to the respective teacher models. The figure shows that the features learned by our approach mimic the representation of its teacher significantly better than the conventional SFT.

Table 4: Comparison of SFT and SFT-KD-Recon for AT distillation method on MRBrainS dataset and 4x acceleration. For SFT, the classification task loss terms are replaced with MSE loss for reconstruction. Our framework which trains in the image domain, ensures better reconstruction fidelity than feature domain learning.

| Dataset | SFT training setup | | | SFT-KD-Recon training setup | | |
| --- | --- | --- | --- | --- | --- | --- |
| | Training done with MSE loss in place of CE/KL loss | | | Training done in image domain with MSE loss | | |
| | Model | HFN/VIF | PSNR/SSIM | Model | HFN/VIF | PSNR/SSIM |
| MRBrainS, 4x | Teacher | 0.4947/0.7379 | 34.06/0.9291 | Teacher | 0.2802/ 0.9146 | 40.05/0.9785 |
| | SF Teacher | 0.2769/0.9223 | 34.06/0.9296 | SFT-KD-Recon Teacher | 0.2769/0.9223 | 40.23/0.9799 |
| | Student | 0.5316/0.6994 | 33.48/0.6994 | Student | 0.3266/0.9037 | 39.49/0.9759 |
| | KD | 0.5534/0.6850 | 32.50/0.9156 | KD | 0.3023/0.9103 | 39.76/0.9769 |
| | SFT | **0.5380/0.7043** | **33.50/0.9205** | SFT-KD-Recon | **0.2816/0.9166** | **40.07/0.9789** |

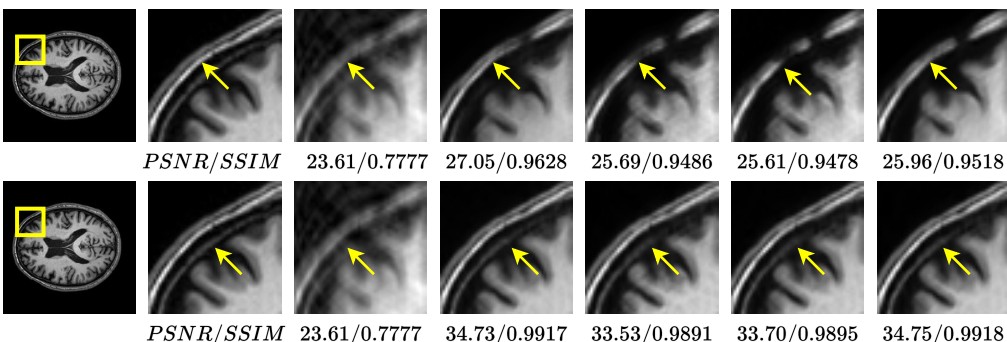

*PSNR/SSIM*     23.61/0.7777     27.05/0.9628     25.69/0.9486     25.61/0.9478     25.96/0.9518

*PSNR/SSIM*     23.61/0.7777     34.73/0.9917     33.53/0.9891     33.70/0.9895     34.75/0.9918

Figure 5: Visual results: Top - SFT training setting (from left to right): target, target inset, ZF, teacher, student, Std. KD, SFT; Bottom - SFT-KD-Recon setting (left to right): target, target inset, ZF, teacher, student, Std. KD, SFT-KD-Recon for MRBrainS dataset with 4x acceleration factor.

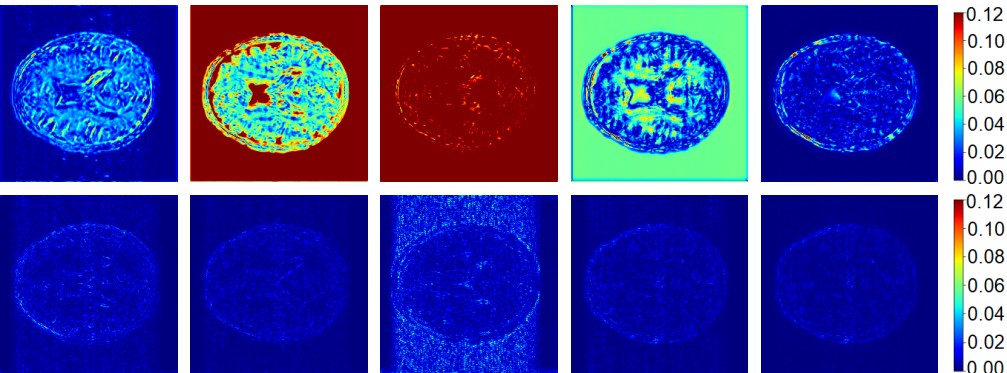

Figure 6: Top - SFT training setting (from left to right): Residue computed between feature attention maps of the distilled student and feature attention maps of teacher from cascade 1 to cascade 5. Bottom - SFT-KD-Recon setting (left to right): Residue computed between distilled student and teacher features from cascade 1 to cascade 5. Note that the distilled student trained in our SFT-KD-Recon setting is able to mimic the teacher's attention maps better.

**Comparison of SFT-KD-Recon with and without random initialization for the student during KD step:** One key difference between the SFT and ours is that the initial knowledge obtained by the student during this mutual learning is not utilized in SFT. We consider this a pre-training phase for the later layers of the student and reuse the initialization of the student during the KD stage. This fine-tuning phase can enable the student to well handle target tasks better than KD, especially for image restoration tasks where fine-image details need to be recovered. As shown in the ablative study (Table 5) with these initializations, the student is performing better than the SFT framework itself.

Table 5: Ablative study comparing SFT-KD-Recon student with random initialization (SFT-KD-Recon (random)) and SFT-KD-Recon student with initialization learned during teacher training stage for AT method on MRBrainS dataset and 4x acceleration in the image domain. Here, our framework, which trains with better initializations, ensures better performance. In general, fine-tuning often outperforms training from scratch because the student during SFT training already has a generous amount of knowledge.

| Evaluation Metrics | Teacher (D5C5) | Student (D3C5) | KD (D3C5) | SFT-KD-Recon (random) (D3C5) | SFT-KD-Recon (D3C5) |
|---|---|---|---|---|---|
| HFN | 0.2802 +/- 0.0319 | 0.3266 +/- 0.02357 | 0.3023 +/- 0.0313 | 0.2882 +/- 0.02705 | 0.2825 +/- 0.02775 |
| MSE | 1.002e-04 +/- 4.626e-05 | 1.137e-04 +/- 4.623e-05 | 1.07e-04 +/- 4.641e-05 | 1.015e-04 +/- 4.510e-05 | 9.967e-05 +/- 4.513e-05 |
| PSNR | 40.05 +/- 2.023 | 39.49 +/- 1.778 | 39.76 +/- 1.899 | 39.99 +/- 1.947 | 40.07 +/- 1.983 |
| SSIM | 0.9785 +/- 0.00655 | 0.9759 +/- 0.005949 | 0.9769 +/- 0.006172 | 0.9785 +/- 0.006112 | 0.9789 +/- 0.0062 |
| VIF | 0.9146 +/- 0.01749 | 0.9037 +/- 0.01972 | 0.9103 +/- 0.01752 | 0.9140 +/- 0.01702 | 0.9141 +/- 0.01799 |

### A.0.3. Versatility of SFT-KD

Among all the KD methods, the SFT-KD-Recon for AT shows the highest performance improvements in all configurations. Using AT, we further analyzed the performance of SFT-KD-Recon with a heterogeneous teacher, wherein we chose DC-UNet (Sun et al., 2019)

as the teacher and D3C5 as the student (different architectures for teacher and student) tabulated in Table 6. This shows that our approach can improve the KD irrespective of whether the teacher's architecture is homogeneous with the student.

Table 6: Ablative study comparing KD and SFT-KD-Recon for AT method on MRBrainS dataset and 4x acceleration by considering DC-UNet as teacher and D3C5 as a student. Despite the architectural mismatch of the student and the teacher, there is visible performance gain of the student using our approach.

| Evaluation Metrics | Teacher | SFT-Teacher | Student | KD | SFT-KD Recon |
|---|---|---|---|---|---|
| PSNR | $40.11 \pm 2.588$ | $\mathbf{40.18 \pm 2.692}$ | $39.47 \pm 1.823$ | $39.77 \pm 1.899$ | $\mathbf{39.89 \pm 1.997}$ |
| SSIM | $0.9777 \pm 0.008232$ | $\mathbf{0.9775 \pm 0.009105}$ | $0.9758 \pm 0.006268$ | $0.9774 \pm 0.006272$ | $\mathbf{0.978 \pm 0.006585}$ |

### A.0.4. COMPARISON WITH OTHER RECONSTRUCTION METHODS

Table 7 illustrates that our model can successfully compress larger networks while performing competitively with other MRI reconstruction networks. The table comparison reveals that our model outperforms three other methods (Sun et al., 2019; Schlemper et al., 2017; Murugesan et al., 2020) and exhibits a competitive Structural Similarity Index Measure of 0.907, which is on par with the best-performing model MAC-ReconNet (Ramanarayanan et al., 2020b) with three times more parameters (0.9114). These observations highlight the importance of model compression for MRI reconstruction.

Table 7: Ablative study comparing SFT-KD-Recon student for AT method on cardiac dataset for 4x and 5x acceleration.

| Reconstruction Method | PSNR/SSIM | | Number of parameters |
|---|---|---|---|
| | **4x** | **5x** | |
| DC-CNN | 32.15/0.9108 | 31.25/0.8964 | 141765 |
| DC-RDN | 31.65/0.9015 | 30.65/0.8844 | 141765 |
| MAC-ReconNet | 32.21/0.9114 | 31.12/0.8943 | 141765 |
| DAGAN | 28.52/0.8410 | 28.02/0.8250 | 3348227 |
| KD-MRI (D3C5) | 31.95/0.9060 | 30.88/0.8879 | 49285 |
| SFT-KD-Recon (D3C5) | 32.03/0.9070 | 30.93/0.8884 | 49285 |

**Comparison of SFT-KD-Recon Teacher training with pre-trained weights and random initializations:** Figure 7 shows validation curves of SFT-KD-Recon teacher (pre-trained on the cardiac dataset and fine-tuned in brain dataset, green curve) and SFT-KD-Recon teacher trained from scratch on brain dataset (red curve). We can see the rapid adaptation of the pre-trained teacher in about 13 epochs taking 4.5 secs against the teacher trained from scratch, taking the number of epochs thrice to converge thereby reducing the computational speed of the training teacher.

**Performance analysis for different compression factors:** We considered different compression factors of the teacher network like D4C5 (33%), D3C5 (62%), and D2C5 (97%) as student networks and tabulated the performance of distilled student networks using conventional KD and proposed approach in Table 8. from the table we note that as the compression rate is very high in D2C5, conventional KD fails due to very small size of

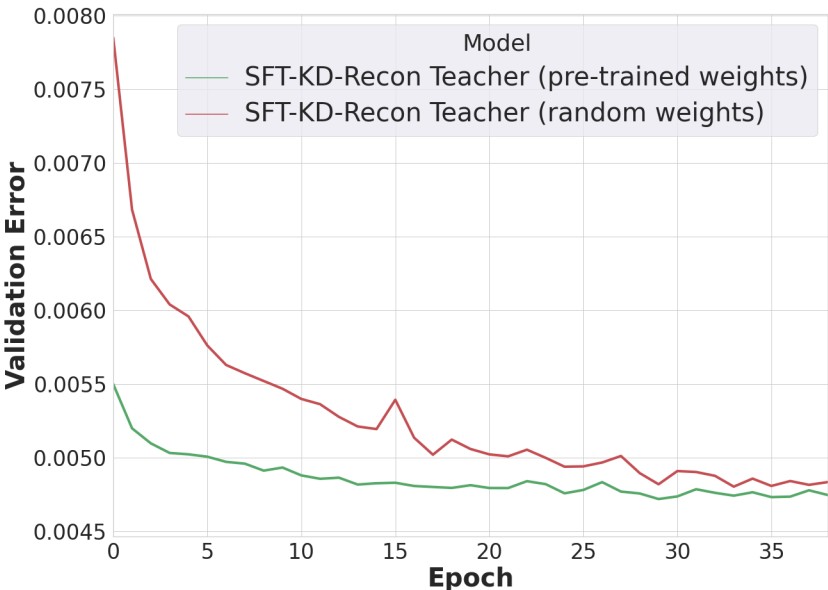

Figure 7: Comparison of the validation loss of SFT-KD-Recon teacher (pre-trained on the cardiac dataset and fine-tuned in brain dataset, green curve) and SFT-KD-Recon teacher trained from scratch on brain dataset (red curve). The figure shows the rapid adaptation of the pre-trained teacher in about 13 epochs, taking 4.5 secs against the teacher trained from scratch, taking the number of epochs thrice to converge.

the network where as the same network distilled with our approach is able to give better performance.

Table 8: Performance analysis for different compression factors (different student models) for MRBrainS Dataset, 4x acceleration factor. Note that the compression factor are 33%, 62% and 97% for D4C5, D3C5, D2C5 respectively.

| MODEL | Parameters | Inference Time (ms) | Original | | KD | | SFT-KD-Recon | |
|---|---|---|---|---|---|---|---|---|
| | | | PSNR/SSIM | HFN/VIF | PSNR/SSIM | HFN/VIF | PSNR/SSIM | HFN/VIF |
| Teacher (D5C5) | 141765 | 17.415 | 40.05/0.9785 | 0.2802/0.9146 | - | - | - | - |
| Student-1 (D4C5) | 95525 (32.6%) | 9.353 | 39.98/0.9785 | 0.2934/0.9113 | 40.19/0.9791 | 0.2774/0.9202 | **40.21/0.9797** | **0.2764/0.9196** |
| Student-2 (D3C5) | 49285 (65.25%) | 6.628 | 39.49/0.9759 | 0.3266/0.9037 | 39.76/0.9769 | 0.3023/0.9103 | **40.07/0.9789** | **0.2825/0.9141** |
| Student-3 (D2C5) | 3045(97.83%) | 3.686 | 39.01/0.9728 | 0.3456/0.8876 | 38.87 /0.9720 | 0.3578/0.8854 | **39.17/0.9739** | **0.3417/0.8929** |

## Appendix B. MRI Super Resolution:

### B.1. MRI Super-Resolution architecture

MRI Super-Resolution involves reconstructing a high-resolution (HR) image from a low-resolution (LR) image. In general, interpolation fails to recover the loss of high-frequency details (fine edges). So, deep learning architectures like VDSR (Kim et al., 2016) were proposed to restore the LR image. VDSR architecture consists of $n$ blocks of convolution

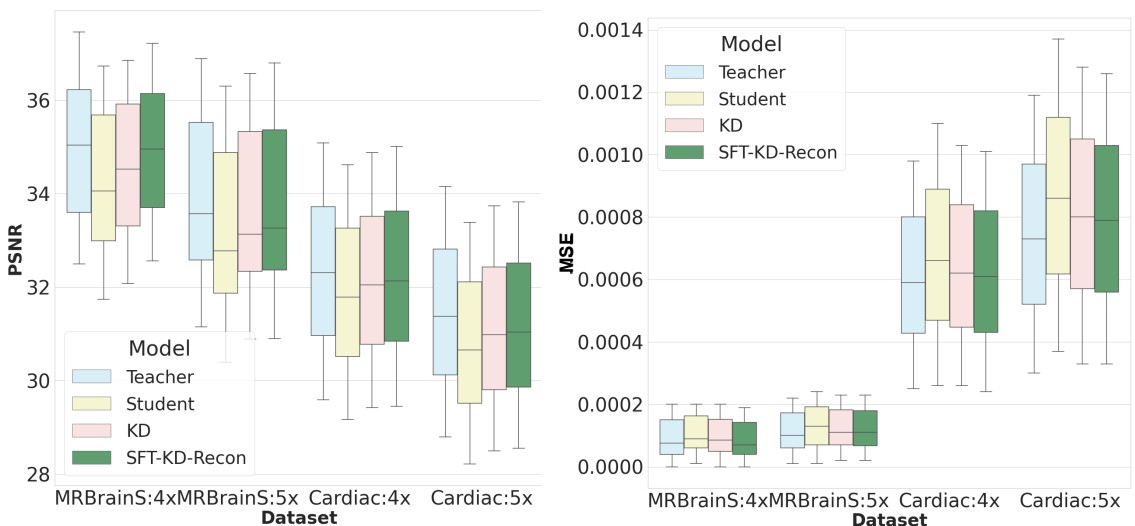

Figure 8: PSNR and MSE Box plots of KD (AT), SFT-KD-Recon with respect to teacher and student across validation data of MRBrainS and cardiac datasets for 4x and 5x acceleration factors. From the plots, it is evident that our approach improves student performance in terms of PSNR also. Our approach (green) has more PSNR, and less mean square error than student and KD

and ReLU with a residual connection between the input and the output. LR image is interpolated to match the dimension of the HR image. We consider VDSR with 12 convolution layers with a residual connection between input and output for the teacher network and 4 convolution layers with a residual connection between input and output for the student network. We evaluate the performance of the student network using our approach and standard approach by considering various baseline KD methods such as AT (Zagoruyko and Komodakis, 2016), Fitnets (Romero et al., 2014), CC (Peng et al., 2019), Neuron selectivity transfer (NST) (Huang and Wang, 2017), Factor transfer (FT) (Kim et al., 2018), Probabilistic knowledge transfer (PKT) (Passalis and Tefas, 2018).

## B.2. Experiments and Results:

### B.2.1. Dataset Description and Evaluation metrics

Dataset Description: **Calgary dataset[Complex-valued]:** The human brain dataset was obtained with 12-channel receiver coil and then combined to fetch a single-coil acquisition. The dataset contains 35 volumes of T1 each of size 256 x 256. We consider the center 110 slices from each volume, which provided 25 volumes (2750 slices) and 10 volumes (1100 slices) for training and validation, respectively.

We use Peak Signal-to-Noise Ratio (PSNR) and Structural Similarity Index (SSIM) metrics as our evaluation metrics.

Table 9: Comparison of our framework with standard KD framework for MRI super-resolution on MRBrainS dataset for 4x acceleration.

| RESOLUTION TASK | | | | | |
|---|---|---|---|---|---|
| | | 4x | | | |
| | | Std-KD | | SFT-KD-Recon | |
| **Dataset** | **Model** | PSNR | **SSIM** | PSNR | SSIM |
| Calgary | Teacher | 30.40+/- 1.345 | **0.8753 +/- 0.01291** | 30.49 +/- 1.262 | **0.8767 +/- 0.01268** |
| | Student | 29.94 +/- 1.423 | **0.8656 +/- 0.01412** | - | - |
| | AT | 29.97 +/- 1.448 | **0.8662 +/- 0.01404** | **30.02 +/- 1.392** | **0.8673 +/- 0.01364** |
| | Fitnets | 29.95 +/- 1.38 | **0.8658 +/- 0.01373** | **30.01 +/- 1.381** | **0.8672 +/- 0.01360** |
| | CC | 29.98 +/- 1.423 | **0.8730 +/- 0.01277** | **29.99 +/- 1.425** | **0.8663 +/- 0.01328** |
| | PKT | 29.95 +/- 1.435 | **0.8659 +/- 0.01381** | **30.01 +/- 1.421** | **0.8676 +/- 0.01349** |
| | factortransfer | 29.99 +/- 1.427 | **0.8665 +/- 0.01332** | **30.04 +/- 1.387** | **0.8677 +/- 0.01360** |
| | NST | 29.98 +/- 1.364 | **0.8662 +/- 0.01374** | **29.99 +/- 1.414** | **0.8663 +/- 0.01382** |

B.2.2. EXPERIMENTS AND RESULTS:

Table 9 shows the quantitative analysis of the distilled student performance of the proposed SFT approach applied to various KD methods. From the table, we observe that SFT-KD performs consistently better than the routine KD approach and the student-friendly teacher performs better than the conventional teacher. In terms of PSNR, the performance drop in the student from the teacher is around 0.46 dB without KD. By using normal KD methods, this drop reduces to only around 0.41 dB. With the proposed SFT-KD, the gap is significantly minimized to 0.36 dB.

