# OpenReview forum: "SFT-KD-Recon: Learning a Student-friendly Teacher for Knowledge Distillation in Magnetic Resonance Image Reconstruction"
_MIDL.io/2023/Conference — MIDL 2023 Poster_

### Official Review · Reviewer_jJYS · 2023-01-27

**Confidence:** 4
**Preliminary Rating:** 2

**Summary:**

This paper is an extended application of the student-friendly teacher training method (NeurIPS’21, https://openreview.net/forum?id=0xs40KGnsq3). Here, the authors apply this spirit to the MRI reconstruction task. The main spirit is jointly training the teacher model with the student model, so that the teacher model can be more effectively adapted during the KD process. The method is validated on two datasets.

**Strengths:**

A strength is similar to their reference NeurIPS paper – the problem may be important in KD.
Overall, the paper is well-organized and well-written.
The method is validated on two datasets and shows some improvement.


**Weaknesses:**

The novelty seems limited. I believe that just replacing the original method with the reconstruction architecture is not so interesting to MIDL readers. Considering that this paper is a direct application of the student-friendly teacher training method (NeurIPS’21), authors should further provide insightful discussion and highlight the challenging part when applying this method to MRI reconstruction to show that it is not a trivial combination/application. Otherwise, readers could not gain enough insights into this paper and may be more interested in the original paper.

Requirement of the student architecture. In the case that student and teacher architectures are heterogeneous, how will the method here perform? The common KD problem does not require the prior of student model, yet, here we should know the student model in advance. Will this limit the application in practice? Authors may need some experiments on this.

What is the KD in Figure 4 standing for? You mention several KD baselines.

A significance test is needed. The improvement is very marginal. Therefore, it is hard to discern whether this training is interesting to the MRI reconstruction community since we should define the student model in advance, which seems not practical. I mean, in previous knowledge distillation, we have a well-trained teacher model, and we can define a shallow student model when we need to compress. Then, we can just perform the KD process. Yet, this method requires the student model during training, i.e., if I have a different compression ratio requirement, I should even retrain the teacher with the new student architecture. If the improvement is very marginal, it is little interesting to developers considering the computation cost.


**Deanonymize Review:**

no

**Paper Type:**

methodological development

**Questions To Address In The Rebuttal:**

See above.

The novelty seems limited. Authors should further provide insightful discussion and highlight the challenging part when applying this method to MRI reconstruction to show that it is not a trivial combination/application.

Requirement of the student architecture. In the case that student and teacher architectures are heterogeneous, how will the method here perform? The common KD problem does not require the prior of student model, yet, here we should know the student model in advance. Will this limit the application in practice? Authors may need some experiments on this.

What is the KD in Figure 4 standing for? You mention several KD baselines.

A significance test is needed. This method requires the student model during training, i.e., if I have a different compression ratio requirement, I should even retrain the teacher with the new student architecture. If the improvement is marginal, it is little interesting to developers considering the computation cost.

---

### Official Review · Reviewer_UEy6 · 2023-01-29

**Confidence:** 4
**Preliminary Rating:** 4
**Recommendation:** Poster

**Summary:**

The paper presents a student-friendly teacher training approach for MRI reconstruction, by combining knowledge distillation (KD) and deep cascaded CNN to let the teacher network be aware of the student network. The teacher network and the student network are jointly trained to make sure the representations of them can be aligned. The experimental results demonstrate that the proposed KD strategy can significantly improve the performance of the existing KD methods.

**Strengths:**

The authors adopted KD to improve the performance of deep cascaded CNN, which is a good strategy for the MRI reconstruction task, and the experimental comparisons with multiple KD methods indicate the superiority of the student-friendly teacher training approach.

**Weaknesses:**

The meaning of using KD for the MRI reconstruction task is not very clear, indeed, the experimental results show the effectiveness of the proposed strategy, but the authors do not compare the proposed method with the existing MRI reconstruction methods, which cannot sufficiently verify the effectiveness of the proposed method. In addition, the proposed SFT-KD-Recon is similar to the SFT (Park et al., 2021), and the innovation in this paper is weak.

**Deanonymize Review:**

no

**Paper Type:**

methodological development

**Questions To Address In The Rebuttal:**

1) Please make it clear that what the difference between SFT-KD-Recon and SFT is.
2) Please make it clear that the meaning of using KD for the MRI reconstruction task.
3) Please compare the proposed method with other MRI reconstruction methods to demonstrate the effectiveness of the proposed method in MRI reconstruction task.

I believe the difference between SFT-KD-Recon and SFT has been clarified, the significance of using KD for MRI reconstruction task has been well explained, and the additional experiments can support the contributions of this paper. Thus, I change the rating from "weak reject" to "weak accept".

---

### Official Review · Reviewer_MNm5 · 2023-02-06

**Confidence:** 3
**Preliminary Rating:** 4
**Recommendation:** Oral, Poster

**Summary:**

This work adapted Student-Friendly Teacher (SFT) training framework (Park et al., 2021) for Magnetic Resonance Image
Reconstruction application. Two datasets have been used to evaluate the proposed method and the performance was measured by PSNR and SSIM. The SFT-KD method showed marginal improvement compared to 5 standard KD frameworks.

**Strengths:**

+ The manuscript was well-written and Figure 2 was well-illustrated. And the detailed explanations in the appendix were appreciated.
+ The experiments were comprehensive, including 2 MRI datasets, 5 standard KD frameworks, and 2 performance metrics.

**Weaknesses:**

- The novelty of the proposed SFT-KD-Recon is very limited compared to the original SFT method.
- The performance improvement SFT-KD-Recon is very marginal compared to the standard KD frameworks in all experiments.

**Deanonymize Review:**

no

**Detailed Comments:**

Please refer to the "Questions To Address In The Rebuttal" Section.

**Paper Type:**

methodological development

**Questions To Address In The Rebuttal:**

1. Limited Novelty
The proposed SFT-KD-Recon is not much different compared to the original SFT. The loss functions were changed from CrossEntropy/KL to L1 for the reconstruction tasks. Please make it clear if any other contribution has been made.

2. Limited Performance Improvement
The improvement only seems marginal in all experiments (<1 for PSNR and <0.01 for SSIM), including but not limited to the comparison between Std-KD and SFT-KD-Recon, SFT-KD with random initialization and SFT-KD with initialization, etc. Please make it clear if the proposed SFT-KD-Recon significantly outperforms standard KD frameworks.

My concerns, especially the concern about limited novelty, have been successfully addressed in the rebuttal.
Therefore, I would like to change my rating from weak reject to weak accept.

---

### Meta-Review · Area_Chair_oy24 · 2023-02-23

**Recommendation:** Accept (Poster)
**Confidence:** 4

**Metareview:**

This work presents a student-friendly teacher (SFT) training approach for MRI reconstruction by using knowledge distillation (KD). The structure and presentation of this paper are clear. Although all reviewers raised concerns on the novelty, these concerns were well dispelled in the rebuttal. This work is valuable in MRI reconstruction  due to its promising ability in knowledge distillation.

Pros:
- KD is a useful strategy for MRI reconstruction
- The experiments of validating this work is convincing
- The proposed method can address heterogeneous cases

All reviewers rated weak reject before the rebuttal due to their main concerns on novelty, but two of three changed their rating to weak accept after the rebuttal. Thanks to the detailed and convincing rebuttal from authors, I recommend to accept this paper.